# Aircraft based Stereographic Reconstruction of 3D Cloud Geometry

Tobias Kölling[1], Tobias Zinner[1], and Bernhard Mayer[1]

[1]Ludwig Maximilians Universität, Meteorologisches Institut, München, Germany

**Correspondence:** T. Kölling (tobias.koelling@physik.uni-muenchen.de)

**Abstract.** This work describes a method to retrieve location and geometry of clouds using RGB images from a video camera on an aircraft and data from the aircraft's navigation system. Opposed to ordinary stereo methods where two cameras with fixed relative position at a certain distance are used to match images taken at the exact same moment, this method uses only a single camera and the aircrafts movement to provide the needed parallax. Advantages of this approach include a relatively simple installation on a (research) aircraft and the possibility to use different image offsets, even larger than the size of the aircraft. Detrimental effects are the evolution of observed clouds during the time offset between two images as well as the background wind. However we will show that some wind information can also be recovered and subsequently used for physics based filtering of outliers. Our method allows the derivation of cloud top geometry which can be used, e.g., to provide location and distance information for other passive cloud remote sensing products. In addition it can also improve retrieval methods by providing cloud geometry information useful for the correction of 3D illumination effects. We show that this method works as intended by comparison to data from a simultaneously operated lidar system. The stereo method provides lower heights than the lidar method, the median difference is $126 \mathrm{~m}$. This behaviour is expected as the lidar method has a lower detection limit (leading to greater cloud top heights for the downward view) while the stereo method also retrieves data points on cloud sides and lower cloud layers (leading to lower cloud heights). Systematic errors across the measurement swath contribute less than $50 \mathrm{~m}$.

## 1 Introduction

As implied by the name of remote sensing, the observer is located at a position different from the observed objects. Accordingly, the location of a cloud is not trivially known in cloud remote sensing applications. Thus, cloud detection, cloud location and cloud geometry are parameters of high importance for all consecutive retrieval products. These parameters themselves govern characteristics like cloud mass or temperature and subsequently thermal radiation budget and thermodynamic phase. Typically passive remote sensing using spectral information is used to retrieve cloud properties including cloud optical thickness, effective droplet radius, thermodynamic phase or liquid water content. However, these methods cannot directly measure the cloud's location. To put the results of such retrieval methods into context, the location must be obtained from another source.

Additional to a missing spatial context, unknown cloud location and geometry are the central reason for uncertainties in microphysical retrievals because of the complex impact of 3D structures on radiative transport (e.g. Várnai and Marshak, 2003; Zinner and Mayer, 2006). The classic method of handling complex, inhomogeneous parts of the atmosphere (e.g. typical

MODIS retrievals) is to exclude these parts from further processing. This of course can severely limit the applicability of such a method. As shown by Ewald (2016) and Ewald et al. (2018) the local cloud surface orientation affects retrieval results. In particular, Ewald (2016) and Ewald et al. (2018) have shown that changes in surface orientation and changes in droplet effective radius produce a very similar spectral response. Thus an independent measurement of cloud surface orientation would very

likely improve retrieval results on droplet effective radius.

As location and geometry information is of such a great importance, a couple of different approaches to get this information can be found. Among these are active methods using lidar or radar. Fielding et al. (2014) and Ewald et al. (2015) show how 3D distributions of droplet sizes and liquid water content of clouds can be obtained by the use of a scanning radar. Ewald et al. (2015) even visually demonstrate the quality of their results by providing simulated images using the retrieved 3D

distributions as input and comparing them to actual photographs. Major downside of this approach is the limited scanning speed. Consequently these methods are especially difficult to employ on fast moving platforms. For this reason, the typical implementations of cloud radar and lidar on aircraft only provide data directly below the aircraft.

Passive methods are often less accurate but can cover much larger observation areas in shorter measurement times. They typically either use spectral features of the signal or use observations from multiple directions. MODIS (Moderate Resolution

Imaging Spectroradiometer) cloud top height for example uses thermal infrared images to derive cloud top brightness temperatures (Strabala et al., 1994). Using assumed cloud emissivity and atmospheric temperature profiles, cloud top heights can be calculated. Várnai and Marshak (2002) used gradients in the MODIS brightness temperature to further classify observed clouds into "illuminated" and "shadowy" clouds. Another spectral approach has been demonstrated amongst others by Fischer et al. (1991) and Zinner et al. (2018) using oxygen absorption features to estimate the travelled distance of the observed light

through the atmosphere. Assuming most of the light gets reflected at or around the cloud surface, this information can be used to calculate the location of the clouds surface.

Other experiments (e.g. Beekmans et al., 2016; Crispel and Roberts, 2018; Romps and Öktem, 2018) use multiple ground based all-sky cameras and apply stereophotogrammetry techniques to georeference cloud fields. Due to the use of multiple cameras, it is possible to capture all images at the same time, so that cloud evolution and motion does not affect the 3D

reconstruction.

Spaceborne stereographic methods have been employed e.g. for the Multi-angle Imaging SpectroRadiometer (MISR) (Moroney et al., 2002) and the Advanced Spaceborne Thermal Emission and Reflection Radiometer (ASTER) (Seiz et al., 2006). MISR features 9 different viewing angles which are captured during 7 minutes of flight time. During the long time period of about one minute between two subsequent images the scene can change substantially. Clouds in particular are transported and

deformed by wind, which adds extra complexity on stereographic retrievals. The method by Moroney et al. (2002) addresses this problem by tracking clouds along all the perspectives and derivation of a coarse wind field at a resolution of about $70 \, \mathrm{km}$. ASTER comes with only two viewing angles but still takes about $64 \, \mathrm{s}$ to complete one image pair. Consequently, the method by Seiz et al. (2006) uses other sources of wind data (e.g. MISR or geostationary satellite data) to correct for cloud motion during the capturing period.

Parts of the introduction refer to the cloud surface, a term which comes with some amount of intuition, but is hard to define in precise terms. This difficulty arises because a cloud has no universally defined boundaries but rather changes gradually between lower and higher concentrations of hydrometeors. Yet, there are many uses for a defined cloud boundary. Horizontal cloud boundary surfaces are commonly denoted as cloud base height and cloud top height, which by their correspondence to the atmospheric temperature profile and subsequently thermal radiation largely effects the energy balance of clouds. Another

such quantity, namely cloud fraction, is often used for example in atmospheric models to improve the parametrization of cloud-radiation interaction. Still, defining a cloud fraction requires to discriminate between areas of clouds and no clouds, introducing vertical cloud boundary surfaces. Stevens et al. (accepted) illustrate what Slingo and Slingo (1988) already said: cloud amount is "a notoriously difficult quantity to determine accurately from observations." Besides the difficulties in defining a thing like the cloud surface, it is a very useful tool to describe how clouds interact with radiation. This in turn allows us to do a little trick:

we define the cloud's surface as the visible boundary of a cloud in 3D space. This may or may not correspond with gradients of microphysical properties, but clearly captures a boundary of interaction between clouds and radiation. This ensures that the chosen surface is relevant, both to improve microphysical retrievals which are based on radiation from a similar spectral region, as well as to use it in investigating cloud-radiation interaction. Additionally, by definition, the cloud surface is located where an image discriminates between cloud and no cloud, which is a perfect fit for the observation with a camera.

In this work, we present a stereographic method which uses 2D images taken from a moving aircraft at different times to find the georeferenced location of points located on the cloud surface facing the observer. This method neither depends on estimates of the atmospheric state nor does it depend on assumptions on the cloud shape. Contrasting to the spaceborne methods, our method only takes $1\,\mathrm{s}$ for one image pair. Due to the relatively low operating altitude of an aircraft compared to a satellite, the observation angle changes rapidly enough to use two successive images without the application of a wind correction method.

As we employ a 2D imager with a wide field of view, each cloud is captured from many different perspectives (up to about 100 different angles, depending on the distance between aircraft and cloud). Due to the high number of viewing angles, it is possible to derive geometry information of partly occluded clouds. Furthermore, this allows to simultaneously derive an estimate of the 3D wind field and use it to improve the retrieval result.

We demonstrate the application of our method to data obtained in the NARVAL-II and NAWDEX field campaigns (Stevens

et al., accepted; Schäfler et al., 2018). In these field campaigns, the hyperspectral imaging system specMACS has been flown on the HALO aircraft (Ewald et al., 2016; Krautstrunk and Giez, 2012). The deployment of specMACS, together with other active and passive instrumentation, aimed at a better understanding of cloud physics including water content, droplet growth, cloud distribution and geometry. The main component of the specMACS system are two hyperspectral line cameras. Depending on the particular measurement purpose, additional imagers are added. The hyperspectral imagers are operating in the wavelength

range of $400-1000\,\mathrm{nm}$ and $1000-2500\,\mathrm{nm}$ at a spectral resolution of a few nanometers. Further details are described by Ewald et al. (2016). During the measurement campaigns discussed in this work, the two sensors were looking in nadir perspective and have been accompanied by a 2D RGB imager with about twice the spatial resolution and field of view. In this work, we focus on data from the 2D imager, because it allows observing the same cloud from different angles.

In Section 2 we briefly explain the measurement setup. Section 3 introduces the 3D reconstruction method and Section 4 presents a verification of our method. For geometric calibration of the camera we use a common approach of analyzing multiple images of a known chessboard pattern to resolve unknown parameters of an analytic distortion model. Nonetheless, as the geometry reconstruction method is very sensitive to calibration errors, we provide a short summary of our calibration process in Appendix A. We used the OpenCV library (Bradski, 2000) for important parts of this work. Details are listed in Appendix B.

## 2 Measurement Setup

During the NARVAL-II and NAWDEX measurement campaigns specMACS was deployed on-board the HALO aircraft. As opposed to Ewald et al. (2016), the cameras have been installed in a nadir looking perspective. The additional 2D imager (Basler acA2040-180kc camera + Kowa LM8HC objective) has been set up to provide a full field-of-view of approximately $70°$ with 2000 by 2000 pixels and data acquisition frequency at 1 Hz. To cope with the varying brightness during and between flights, the camera's internal exposure control system has been used.

Additionally, the WALES lidar system (Wirth et al., 2009), the HALO Microwave Package HAMP (Mech et al., 2014), the Spectral Modular Airborne Radiation measurement sysTem SMART (Wendisch et al., 2001) and an AVAPS dropsonde system (Hock and Franklin, 1999) was part of the campaign specific aircraft instrumentation. The WALES instrument is able to provide an accurate cloud top height and allows to directly validate our stereo method as described in section 4.

## 3 3D reconstruction

The goal of our 3D reconstruction method is to find georeferenced points which are part of a cloud surface at a specific time in an automated manner. Input data are geometrically calibrated images from a 2D camera fixed to the aircraft. As the aircraft flies, pictures taken at successive points in time show the same clouds from different perspectives. A schematic of this geometry is shown in Fig. 1. The geometric calibration of the camera and the rigid mounting on the aircraft allows to associate each sensor pixel with a viewing direction in the aircraft's frame of reference. The orientation of the camera with respect to the aircraft's frame of reference has been determined by aligning images taken on multiple flights to landmarks also visible in satellite images. Using the aircraft's navigation system, all relevant distances and directions can be transformed into a geocentric reference frame in which most of the following calculations are performed. The reconstruction method contains several constants which are tuned to optimize its performance. Their values are listed in table 1.

In order to perform a stereo positioning, a location on a cloud must be identified in multiple successive images. A location outside of a cloud is invisible to the camera, as it contains clear air, which barely interacts with radiation in the observed spectral range. Locations enclosed by the cloud surface do not produce strong contrasts in the image, as the observed radiation is likely scattered again before reaching the sensor. Thus, a visible contrast on a cloud is very likely originating from a location on or close to the cloud surface as defined in the introduction. This method starts by identifying such contrasts. If such a contrast is only present in one direction of the image (basically, we observe a line), this pattern is not suitable for tracking to the next

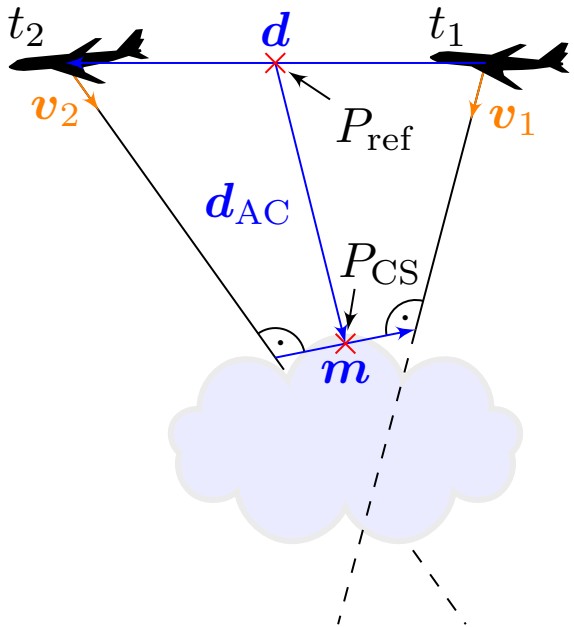

**Figure 1.** Schematic drawing of the stereographic geometry. Images of clouds are taken at two different times from a fast moving aircraft. Using aircraft location and viewing geometry, a point $P_{CS}$ on the clouds surface can be calculated. Note that the drawing is not to scale: $\boldsymbol{d}$ is typically around 200 m, $\boldsymbol{d}_{AC}$ in the order of 5 km and $\boldsymbol{m}$ is a description of mis-pointing and in the order of only a few meters.

image due to the aperture problem (Wallach, 1935). We thus search one image for pixels of which the surroundings show a strong contrast in two independent directions. This corresponds to two large eigenvalues ($\lambda_1$ and $\lambda_2$) of the Hessian matrix of the image intensity. This approach has already been formulated by Shi and Tomasi (1994): interesting points are defined as points with $\min(\lambda_1, \lambda_2) > \lambda$ with $\lambda$ being some threshold. We use a slightly different variant and interpret $\min(\lambda_1, \lambda_2)$ as a quality measure for each pixel. In order to obtain a more homogeneous distribution of tracking points over the image, candidate points are sorted by quality. Points which have better candidates at a distance of less then $r_{\min}$ are removed from the
5 list and the remaining best $N_{\text{points}}$ are taken. For these initial points, matches in the following image are sought using the optical flow algorithm described by Lucas and Kanade (1981). In particular, we use a pyramidal implementation of this algorithm as introduced by Bouguet (2000). If no match can be found, the point is rejected.

  The locations of the two matching pixels define the viewing directions $\boldsymbol{v}_1$ and $\boldsymbol{v}_2$ in Fig. 1. The distance travelled by the aircraft between two images is indicated by $\boldsymbol{d}$. Under the assumption that the aircraft travels much faster than the observed
10 clouds, an equation system for the position of the point on the cloud's surface $P_{CS}$ can be found. In principle, $P_{CS}$ is located at the intersection of the two viewing rays along $\boldsymbol{v}_1$ and $\boldsymbol{v}_2$, but as opposed to 2D space in 3D space there is not necessarily an intersection, especially in presence of inevitable alignment errors. We relax this condition by searching for the shortest distance between the viewing rays. The shortest distance between two lines can be found by introducing a line segment which

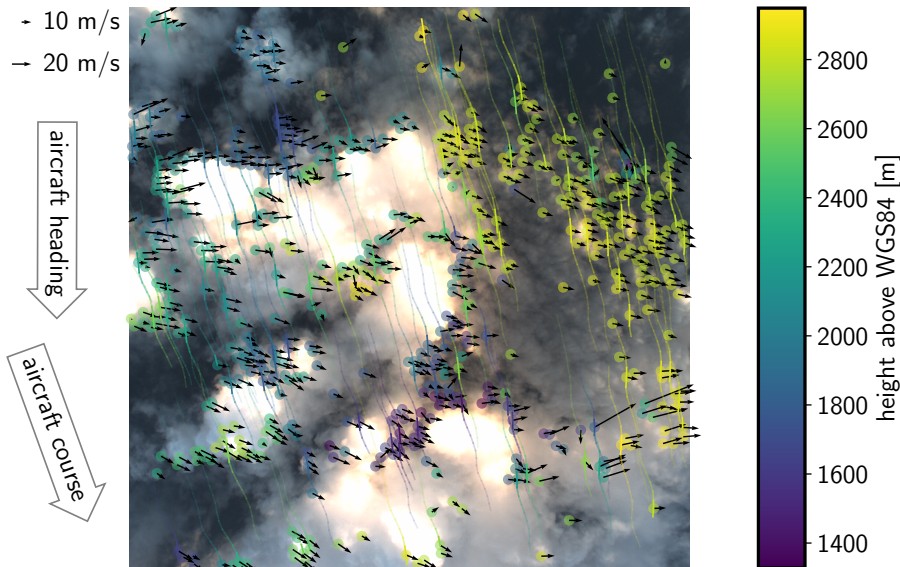

**Figure 2.** Image point tracking, every line in this image represents a cloud feature which has been tracked along up to 30 images. The images used have been taken on the NAWDEX flight RF07 (2016-10-06 09:32:15 UTC, location indicated in Figure 7) in an interval of 1 s. Transparency of the tracks indicates time difference to the image. Color indicates retrieved height above WGS84, revealing that the larger clouds on the left belong to a lower layer than the thin clouds on the right. The arrows indicate estimated cloud movement. Due to the wind speed at the aircraft location, its course differs significantly from the heading and the tracks are tilted accordingly. The number of points shown has been reduced to include at maximum 1 point per 20 px radius in the image. Tracks are only shown for every 5th point.

is perpendicular to both lines. This is the mis-pointing vector $\boldsymbol{m}$. The point on the cloud's surface $P_{\text{CS}}$ is now defined at the center of this vector. If for further processing a single point for the observer location is needed, the point $P_{\text{ref}}$ at the center of both aircraft locations is used.

This way, many points potentially located on a cloud's surface are found. Still, these points contain a number of false correspondences between two images. During turbulent parts of the flight, errors in synchronization between aircraft navigation system and camera will lead to errors in calculated viewing directions. To reject these errors, a set of filtering criteria is applied (the threshold values can be found in Tab. 1). Based on features of a single $P_{\text{CS}}$, the following points are removed:

- $P_{\text{CS}}$ position is behind the camera or below ground

- absolute mis-pointing $|\boldsymbol{m}| > m_{\text{abs}}$

- relative mis-pointing $|\boldsymbol{m}|/|\boldsymbol{d}_{\text{AC}}| > m_{\text{rel}}$

**Table 1.** Filter thresholds

| name | value |
| --- | --- |
| $N_{\text{points}}$ | 1000 |
| $r_{\text{min}}$ | 5 px |
| $m_{\text{abs}}$ | 20 m |
| $m_{\text{rel}}$ | $1.5 \times 10^{-3}$ |
| $v_{\text{jump}}$ | 3 |
| $N_{\text{min}}$ | 5 |
| $d_{\text{abs}}$ | 250 m |
| $d_{\text{rel}}$ | $7 \times 10^{-2}$ |

Figure 2 shows long tracks corresponding to a location on the cloud surface. These tracks follow the relative cloud position through up to 30 captured images. The tracks are generated from image pairs by repeated tracking steps originating at the $t_2$ pixel position of the previous image pair. Using these tracks, additional physics based filtering criteria can be defined.

Each of these tracks contains many $P_{\text{CS}}$ points which should all describe the same part of the cloud. As clouds move with the wind, the points $P_{\text{CS}}$ do not necessarily have to refer to the same geocentric location, but should be transported with the local cloud motion. For successfully tracked points, it can indeed be observed that the displacement of the $P_{\text{CS}}$ points in a 3D geocentric coordinate system roughly follows a preferred direction instead of jumping around randomly, which would be expected if the apparent movement would just be caused by measurement errors. The arrows in Figure 2 show the average movement of the $P_{\text{CS}}$ of each track, reprojected into camera coordinates.

For the observation period (up to 30 s) it is assumed that the wind moves parts of a cloud on almost straight lines at a relatively constant velocity (which may be different for different parts of the cloud). Then, sets of $P_{\text{CS}}$ can be filtered for unphysical movements. The filtering criteria are

- *velocity jumps*: the fraction of maximum to median velocity of a track must be less than $v_{\text{jump}}$

- *count*: the number of calculated $P_{\text{CS}}$ in a track must be above a given minimum $N_{\text{min}}$

- *distance uncertainty*: the distance $\boldsymbol{d}_{\text{AC}}$ between aircraft and cloud may not vary more than $d_{\text{abs}}$ or the relative distance variation with respect to the average distance of a track must be less than $d_{\text{rel}}$

During measurements close to the equator, typical during the NARVAL-II campaign, the sun is frequently located close to the zenith. In this case, specular reflection of the sunlight at the sea surface produces bright spots, known as sunglint and illustrated in Figure 3. Due to waves on the ocean surface, these regions of the image also produce strong contrasts. It turns out that such contrasts are preferred by the Shi and Tomasi algorithm for feature selection, but are useless in order to estimate the cloud surface geometry. To prevent the algorithm from tracking these points, the image area in which bright sunglint is to

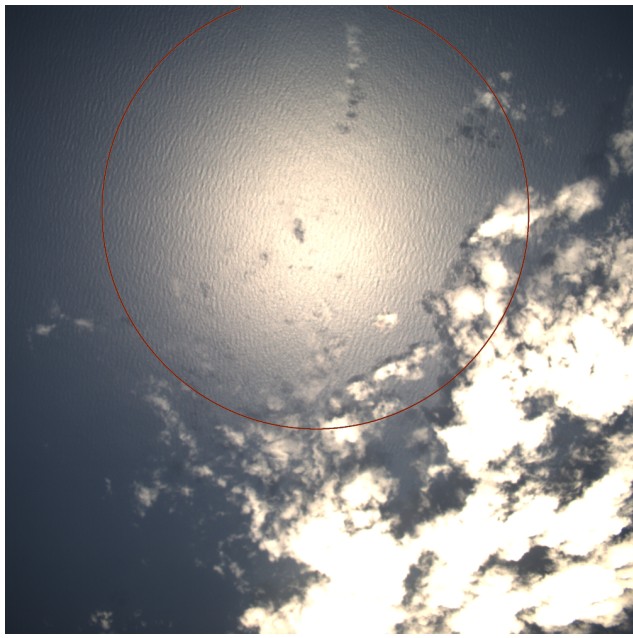

**Figure 3.** At low latitudes, close to the local noon as on the NARVAL-II flight RF07 (2016-08-19 15:06:13 UTC), the specular reflection of the sun on the ocean surface (sunglint) produces bright spots and high contrasts on the waves tails. While the bright spots can visually hide clouds, the contrasts create useless initial tracking points. The latter are mitigated by calculating the region of a potential sunglint (shown as red contour) and masking that region before the images are processed.

be expected is estimated using the bidirectional reflectance distribution function (BRDF) by Cox and Munk (1954) included in the libRadtran package (Mayer and Kylling, 2005; Emde et al., 2016). The resulting area (indicated by a red line in Fig. 3) is masked out of all images before any tracking is performed. Masking out such a large area from the camera image seems to be a wasteful approach. In fact, this is acceptable: due to the large viewing angle of the camera, all masked-out clouds are almost certainly visible at a different time in another part of the image. Therefore, these clouds can still be tracked using parts of the sensor which are not affected by sunglint, even if a large part of the sensor is obstructed by sunglint.

After filtering, a final mean cloud surface point $\bar{P}_{CS}$ is derived from each track as the centroid of all contributing cloud surface points. The collection of all $\bar{P}_{CS}$ form a point cloud in a Cartesian 3D reference coordinate frame which is defined relative to a point on the earth's surface (Figure 4). This point cloud can be used on its own, serve as a reference for other distance measurement techniques (e.g. oxygen absorption methods (Zinner et al., 2018), deriving distances by a method according to Barker et al. (2011)) or allow for a 3D surface reconstruction.

A precise camera calibration (relative viewing angles in the order of $0.01°$) is crucial to this method, which can be achieved by the calibration process as described in Appendix A. A permanent time-synchronization between the aircraft position sensors and the cameras, accurate at the order of tens of milliseconds is indispensable as well. It should be noted that this does involve time stamping each individual image to cope with inter-frame jitter as well as disabling any image stabilization inside the

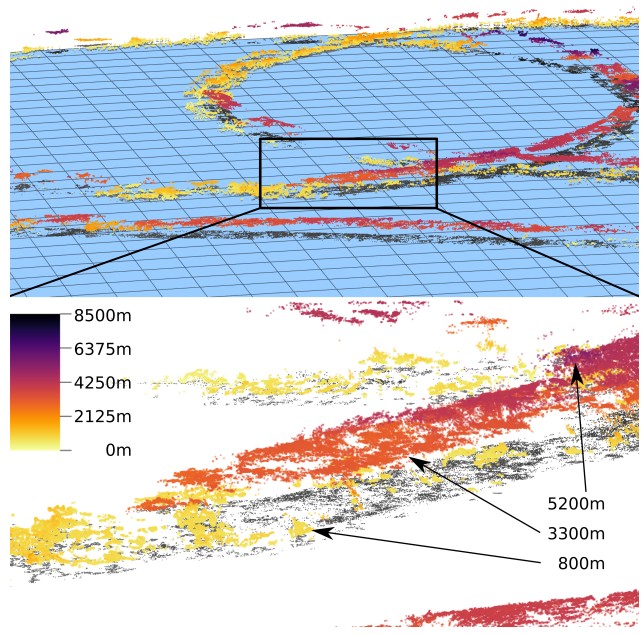

**Figure 4.** The collection of all $\bar{P}_\mathrm{CS}$ form a point cloud. Here, a scene from the second half of the NARVAL-II flight RF07 is shown. The colors indicate the point's height above the WGS84 reference ellipsoid (indicated as blue surface). Below, a part of the scene is shown magnified, displaying two main cloud layers: one at about $800\,\mathrm{m}$ in yellow and the other at about $3200\,\mathrm{m}$ in orange. On the right, a small patch of even higher clouds is visible at $5200\,\mathrm{m}$. The gray dots are a projection of the points onto the surface to improve visual perception.

camera. As this involves generating data which is only available during the measurement, this must be considered prior to the system deployment. For the system described in this work, we used the network time protocol (Mills et al., 2010) with an update interval of $5\,\mathrm{min}$.

## 4 Verification

### 4.1 Across track stability and signal spread

Errors in the sensor calibration could lead to systematic errors in the retrieved cloud height with respect to lateral horizontal distance relative to the aircraft (perpendicular to flight track). In order to assess these errors, data from a stratiform cloud deck observed between 09:01:25 and 09:09:00 UTC during NAWDEX flight RF11 on 2016 Oct 14 has been sorted by the average across track pixel position. While the cloud deck features a lot of small scale variation, it is expected to be almost horizontal on average. Note that as the orientation of the camera with respect to the aircraft has been determined independently using landmarks, deviations from the assumption of a horizontal cloud deck should be visible in the corresponding data and are

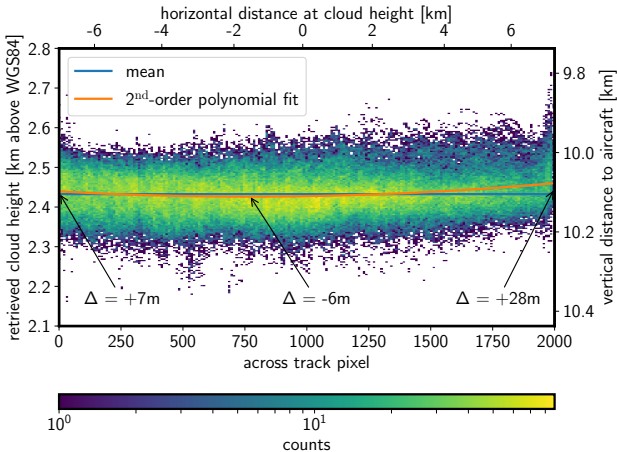

**Figure 5.** In the time from 09:01:25 to 09:09:00 UTC during NAWDEX flight RF11 on 2016 Oct 14, a stratiform cloud deck has been observed. The parabolic fit shows that a small systematic variation can be found beneath the noise (which is due to small scale cloud height variations and measurement uncertainties). Compared to the overall dimensions of the observed cloud ($\approx 14$ km) and the uncertainty of the method, these variations are small. It may still be noted, that data from the edges of the sensor ($\approx 50$ px on each side) should be taken with care.

counted as additional retrieval uncertainty in this analysis. During the investigated time frame, 260360 data points have been collected using the stereo method. The vertical standard deviation of all points is $47.3$ m, which includes small scale cloud height variation and measurement error. Figure 5 shows a 2D histogram of all collected data points. From visual inspection of the histogram, apart from about $50$ px at the sensor's borders, no significant trend can be observed. To further investigate the errors, a $2^{\text{nd}}$-order polynomial has been fitted to the retrieved heights. This polynomial is chosen to cover the most likely effect of sensor mis-alignment which should contribute to a linear term and distortions in the optical path which should contribute to a quadratic term. The difference between left and right side of the sensor of $21$ m corresponds to less than $0.1°$ of absolute camera misalignment and the curvature of the fit is also small compared to the overall dimensions of the observed clouds.

### 4.2 Lidar comparison

Cloud top height information derived from the WALES lidar (Wirth et al., 2009) is used to verify the bias of the described method. While the stereo method provides $P_{\text{CS}}$ at arbitrary positions in space, the lidar data is defined on a fixed grid ("curtain") beneath the aircraft. To match lidar measurements to related stereo data points, we collect all stereo points which are horizontally close to a lidar measurement. This can be accomplished by defining a vertical cylinder around the lidar beam with $150$ m radius. Every stereo derived point which falls into this cylinder with a time difference of less than $10$ s is considered as stereo point related to the lidar measurement. As the (almost) nadir pointing lidar is observing cloud top heights only, we use the highest stereo point inside the collection cylinder. The size of the cylinder is rather arbitrary but the particular choice has reasons: the aircraft moves at a speed of approximately $200$ m/s and the data of the lidar system is available at $1$ Hz

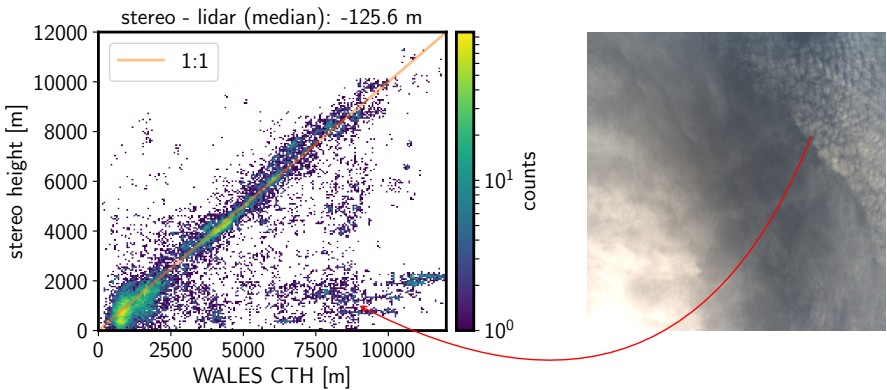

**Figure 6.** Comparison of cloud top height, measured with the WALES lidar and the stereo method. The most prominent outliers, present in the region of high lidar CTH and low stereo height can be attributed to thin, mostly transparent cirrus layers and cumulus clouds below, illustrated by a scene from NAWDEX RF10 (2016-10-13 10:32:10 UTC). While the lidar detects the ice clouds, the stereo method retrieves the height of the cumulus layer below.

and averaged over this period. Any comparison between both systems should therefore be in the order of 200 m horizontal resolution. Furthermore, data derived from the stereo method is only available where the method is confident that it worked. Thus not every lidar data point has a corresponding stereo data point. Increasing the size of the cylinder increases the count of data pairs, but also increases false correspondences. The general picture however remains unchanged.

Figure 6 compares the measured cloud top height from the WALES lidar and the stereo method, visually showing a good agreement. However its quantification in an automated manner and without manual (potentially biased) filtering proves to be difficult. Part of this difficulty is due to the cloud fraction problem, which is explained by (Stevens et al., accepted), basically stating that different measurement methods or resolutions will always detect different clouds. This is also indicated in Figure 6 on the right: the stereo method detects the lower cumulus cloud layer due to larger contrasts while the lidar observes the higher cirrus layer, leading to wrong cloud height correspondences while both methods are supposedly correct. Filtering the data for high lidar cloud top height and low stereo height, reveals that the lower right part of the comparison can be attributed almost exclusively to similar scenes. Further comparison difficulties arise from collecting corresponding stereo points out of a volume which might in fact include multiple (small) clouds. Considering all these sources of inconsistency, only a very conservative estimate of the deviation of lidar and stereo values can be derived from this unfiltered comparison. The median bias between lidar and stereo method is approximately 126 m for all compared flights, indicating lower heights for the stereo method. As the lidar detects cloud top heights with high sensitivity and the stereo method relies on image contrast which is predominantly present at cloud sides, this direction is expected.

Further manual filtering indicates that the real median offset is likely in the order of 50 m to 80 m, however this cannot be shown reliably. Quantifying the spread between lidar and stereo method yields no meaningful results for the same reasons.

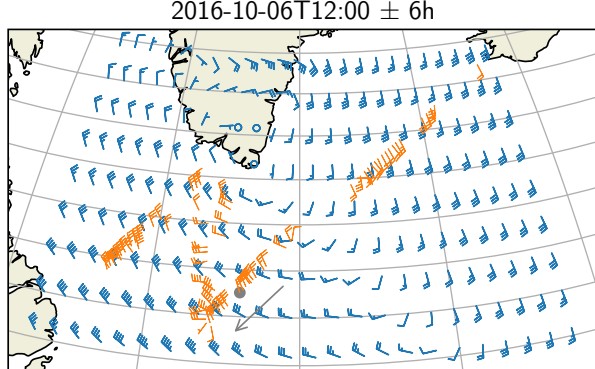

**Figure 7.** Horizontal wind at about 2000 m above ground. Comparison between ECMWF reanalysis (blue) and stereo derived wind (orange). Comparing grid points with co-located stereo data, the mean horizontal wind magnitude is 15.1 m/s in ECMWF and 13.4 m/s in the stereo dataset. This amounts to a difference of $1.7\pm4.5$ m/s in magnitude and $6.0\pm33°$ in direction. The shown deviations are standard deviations over all grid points with co-located data. The gray dot and arrow mark the location and flight course corresponding to Figure 2.

## 4.3 Wind data comparison

An important criteria that we use to identify reliable tracking points is based on the assumption that the observed movement of the points can be explained by a smooth transport due to a background wind field. The thresholds for this test are very tolerant, so the requirements for the accuracy of the retrieved wind field are rather low. However, a clear positive correlation between the observed point motion and the actual background wind would underpin this assumption substantially. In order to do so, we compare the stereo wind against an ECMWF reanalysis in a layer in which many stereo points have been found.

In the following, the displacement vectors of every track have been binned in time intervals of 1 minute along the flight track and 200 m bin in the vertical. To reduce the amount of outliers, bins with less than 100 entries have been discarded. Inside the bins, the upper and lower 20% of the wind vectors (counted by magnitude) have been dropped. All remaining data has been averaged to produce one mean wind vector per bin. In Figure 7 the horizontal component of these vectors is compared to ECMWF reanalysis data at about 2000 m above ground with horizontal sampling of 0.5°. The comparison shows overall good agreement according to our goal to consider a quantity in the stereo matching process which roughly behaves like the wind. The general features of wind direction and magnitude are captured. Deviations may originate from multiple sources including the time difference between reanalysis and measurement, representativity errors and uncertainties of the measurement principle. These results corroborate the assumption that the observed point motion is related to the background wind and filtering criteria based on this assumption can be applied.

## 5 Conclusions

The 3D cloud geometry reconstruction method described in this work is able to produce an accurate set of reference points on the observed surface of clouds. This has been verified by comparison to nadir pointing active remote sensing. Using data from the observation of a stratiform cloud field, we could verify that no significant systematic errors are introduced by looking in off-nadir directions. Even for sunglint conditions cloud top heights can be derived: as clouds move through the image while the sunglint stays relatively stable, we can choose to observe clouds when they are in unobstructed parts of the sensor. Because of the wide field-of-view of the sensor, there are always viewing directions to each cloud which are not affected by the sunglint.

As a visible contrast suited for point matching is a central requirement of the method, it is able to provide positional information at many but not all points of a cloud. Especially flat cloud tops can show very little contrast and are hard to analyse using our method. In future, we will integrate other position datasets like the distance measurement technique using $O_2A$-absorption as described by Zinner et al. (2018) which is expected to work best in these situations. In combining multiple datasets, the low bias and angular variability of the stereo method can even help to improve uncertainties of other methods.

While the wind information derived as part of the stereo method constituted a byproduct of this work the results look promising. After some further investigations about its quality and possibly additional filtering, this product might further add valuable information to the campaign dataset.

During the development of this method, it became clear that a precise camera calibration (relative viewing angles in the order of $0.01°$) is crucial to this method. A permanent time-synchronization between the aircraft position sensors and the cameras, accurate at the order of tens of milliseconds is indispensable as well. It should be noted that this does involve time stamping each individual image to cope with inter-frame jitter as well as disabling any image stabilization inside the camera. For upcoming measurement campaigns, improvements may be achieved by optimizing the automatic exposure of the camera for bright cloud surfaces instead of relying on the built-in exposure system. Furthermore, it would be useful to re-investigate the proposed method with a camera system operating in the near-infrared which would most likely profit from higher image contrasts due to lower Rayleigh scattering in this spectral region.

*Data availability.* specMACS data is available at https://macsserver.physik.uni-muenchen.de after requesting a personal account.

## Appendix A: Geometric Camera Calibration

As the distance between aircraft and observed clouds is typically much larger than the flight distance between two images, the 3D reconstruction method relies on precise measurements of camera viewing angles. To allow analysis as presented in this paper, frame rates of about $1$ Hz are required. At that frame rate, a change in distance between cloud and aircraft of $100$ m at a distance of $10$ km results in approximately $0.01°$ difference of the relative viewing angle or about $1/3$ px. Consequently, achieving accuracies in the order of $100$ m or below requires both, to average over many measurements in order to get sub-pixel

accuracy and to remove any systematic error in the geometric calibration to less than $1/3$ px. This is only possible if distortions in the cameras optical path can be understood and corrected.

We use methods provided by the OpenCV library (Bradski, 2000) to perform the geometric camera calibration, our notation is chosen accordingly. Geometric camera calibration is done by defining a parameterized model which describes how points in world coordinates are projected onto the image plane including all distortions along the optical path. Generally, such a model includes extrinsic parameters which describe the location and rotation of the camera in world space and intrinsic parameters which describe processes inside the camera's optical path. Extrinsic parameters can differ between each captured image while intrinsic parameters are constant as long as the optical path of the camera is not modified. After evaluation of various options for the camera model, we decided to use the following:

$$\begin{pmatrix} x \\ y \\ z \end{pmatrix} = R \begin{pmatrix} X \\ Y \\ Z \end{pmatrix} + \boldsymbol{t} \tag{A1}$$

$$x' = x/z \tag{A2}$$

$$y' = y/z \tag{A3}$$

Where $X, Y, Z$ are the world coordinates of the observed object, $R$ and $\boldsymbol{t}$ the rotation and translation from world coordinates in camera centric coordinates and $x, y, z$ are the object location in camera coordinates. $x'$ and $y'$ are the projection of the object points onto a plane at unit distance in front of the camera. The distortion induced by the lenses and the window in front of the camera is accounted for by adjusting $x'$ and $y'$ to $x''$ and $y''$:

$$r^2 = x'^2 + y'^2 \tag{A4}$$

$$x'' = x'(1 + k_1 r^2 + k_2 r^4 + k_3 r^6) + s_1 r^2 + s_2 r^4 \tag{A5}$$

$$y'' = y'(1 + k_1 r^2 + k_2 r^4 + k_3 r^6) + s_3 r^2 + s_4 r^4 \tag{A6}$$

Here $k_1$ to $k_3$ describe radial lens distortion and $s_1$ to $s_4$ add a small directed component according to the thin prism model. During evaluation of other options provided by OpenCV, no significant improvement of the calibration result was found using more parameters. Finally, the pixel coordinates can be calculated by a linear transformation (which is often called "camera matrix"):

$$u = f_x x'' + c_x \tag{A7}$$

$$v = f_y y'' + c_y \tag{A8}$$

Here, $f_x$ and $f_y$ describe the focal lengths and $c_x$ and $c_y$ describe the principal point of the optical system. In this model, there are 6 extrinsic parameters (rotation matrix $R$ and displacement vector $\boldsymbol{t}$) and 11 intrinsic parameters ($k_1...k_3, s_1...s_4, f_x, f_y, c_x, c_y$).

We use the well-known chessboard calibration method to calibrate this model which is based on Zhang (2000). The basic idea is to relate a known arrangement of points in 3D world space to their corresponding locations on the 2D image plane by a model as described above and solve for the parameters by fitting it to a set of sample images. The internal corners of

a rectangular chessboard provide a good set of such points as they are defined at intersections of easily and automatically recognizable straight lines. Furthermore, the intersection of two lines can be determined to sub-pixel accuracy which improves the calibration performance substantially. While the extrinsic parameters have to be fitted independently for every image, the intrinsic parameters must be the same for each image and can be determined reliably if enough sample images covering the whole sensor are considered.

To evaluate the success of the calibration, the reprojection error can be used as a first quality measure. The reprojection error is defined as the difference of the calculated pixel position using the calibrated projection model and the measured pixel position on the image sensor. To calculate the pixel position, the extrinsic parameters have to be known, thus the images which have been used for calibration are used to calculate the reprojection error as well. This makes this test susceptible to falsely return good results due to an overfitted model. We use many more images (of which each provide multiple constraints to the fit) than parameters to counter this issue and have validated the stereo method (which includes the calibration) against other sensors to ensure that the calibration is indeed of good quality. Nonetheless, a high reprojection error would indicate a problem in the chosen camera model.

We have taken 62 images of a 9 by 6 squares chessboard pattern with $65 \, \mathrm{mm}$ by $65 \, \mathrm{mm}$ square size on an aluminium composite panel using the system assembled in aircraft configuration. The images have been taken such that the chessboard corners are spread over the whole sensor area, Figure A1 shows the pixel locations of all captured chessboard corners. After previous experiments with calibration targets made from paper and cardboard, it became clear that small ripples which inevitably appeared on the cardboard targets render the reprojection errors unusably high. Using the rigid aluminium composite material for the calibration target let the average reprojection error drop by an order of magnitude to a very low value of approximately $0.15$ pixels, which should be enough to reduce systematic errors across the camera to less than $50 \, \mathrm{m}$. The per pixel reprojection error is shown in Figure A2 for every chessboard corner captured.

The calibration used for this work has been performed using gray scale versions of the captured chessboard images. To assess effects of spectral aberrations, the procedure has been repeated separately for each channel. A comparison shows that observed viewing angle differences vary up to the order of $1 - 2$‰ when switching between different calibration data. For the example used in the beginning of this appendix, this translates into cloud height differences of about $10 \, \mathrm{m}$, which is considerably lower than the total errors achievable by this calibration method. Note that besides the same order of magnitude, these effects are not able to explain the curvature in the analysis of Section 4.1. Due to effectively using fewer pixels when doing the camera calibration procedure on a single channel image, the reprojection error is increased accordingly. For these reasons, and in order to facilitate data handling, only one set of calibration data is used. Effects of spectral aberrations within one color channel have not been assessed, but are assumed to be smaller than effects between color channels.

## Appendix B: OpenCV usage

The image processing of this work has been done with help of the OpenCV library (Bradski, 2000). The most important functions used are:

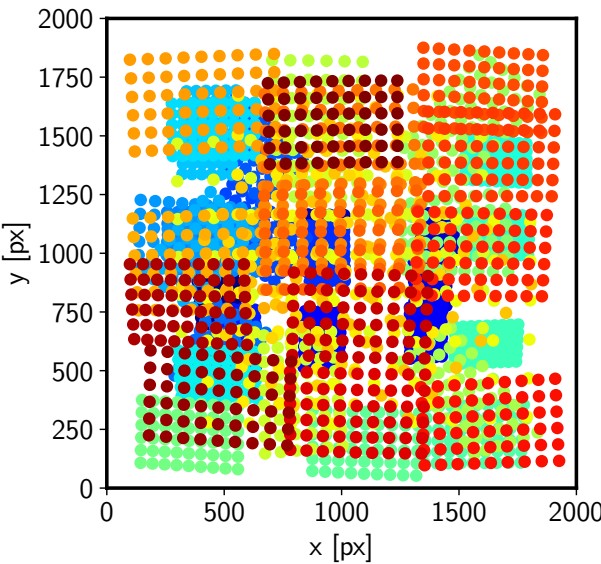

**Figure A1.** Locations of each chessboard corner during calibration on the image sensor plane.

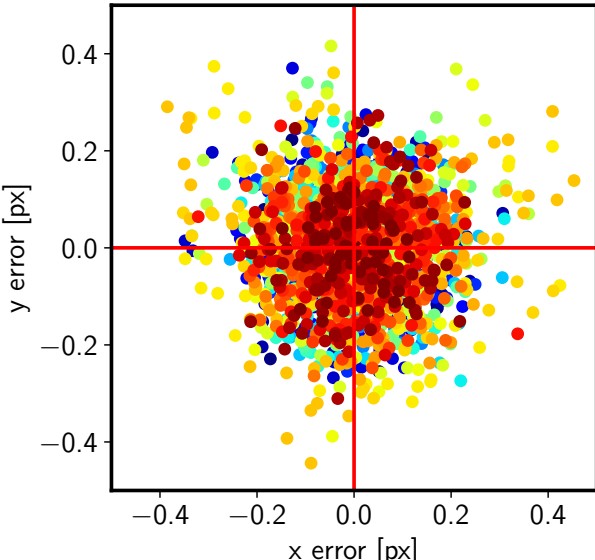

**Figure A2.** Reprojection error: Difference between the actual position of the chessboard corners and the calculated positions of the chessboard corners after applying the camera calibration. The average reprojection error is about $0.15$ pixels. Note that the actual chessboard corner locations can be found far into sub-pixel accuracy by following the lines along the edges of the squares.

- `calibrateCamera` to calculate the camera calibration coefficients from a set of chessboard images

- `goodFeaturesToTrack` to find pixels which are most likely good candidates to be identified in the following image

- `calcOpticalFlowPyrLK` to find the corresponding pixel in the following image

5 *Author contributions.* T. Kölling was in charge of the presented measurements, developed the stereo reconstruction method and prepared the manuscript. T. Zinner and B. Mayer prepared the field campaigns, provided valuable input during the development of the method and contributed to the final version of the manuscript.

*Competing interests.* The authors declare that they have no conflict of interest.

*Acknowledgements.* This work was supported by the DFG (Deutsche Forschungsgemeinschaft, German Research Foundation) through the Project 264269520 "Neue Sichtweisen auf die Aerosol-Wolken-Strahlungs-Wechselwirkung mittels polarimetrischer und hyper-spektraler
5 Messungen". This work was supported by the Max Planck Society, and the DFG (Deutsche Forschungsgemeinschaft, German Research Foundation) through the HALO Priority Program SPP 1294 "Atmospheric and Earth System Research with the Research Aircraft HALO (High Altitude and Long Range Research Aircraft)". The authors thank Hans Grob for his help during the measurements.

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
