# Peer review of "Aircraft based Stereographic Reconstruction of 3D Cloud Geometry"

_Atmospheric Measurement Techniques, 2018_

## Referee Comment (RC1) · Anonymous Referee #1 · 29 Oct 2018

**General Comments:**

The submitted manuscript deals with the 3-D reconstruction of clouds via the structure-from-motion technique using image data obtained from a downward-looking camera installed at a research aircraft. The goal is to provide 3-D cloud top geometry information and geolocation that can be used to improve retrievals by other remote sensing methods, e.g. derivation of cloud droplet radii from spaceborne hyper-spectral imagery. While airborne observation of clouds is a costly enterprise and delivers data only for the flight period, it provides a quite complete and still rare view on the cloud top geometry.

The article describes in detail the methodology and evaluation of the proposed airborne reconstruction, including camera calibration, feature tracking and 3-D reconstruction. Besides an empirical evaluation with an onboard lidar system, the article discusses related challenges of such an approach, such as synchronization with the aircrafts navigation system or the effect of cloud evolution and motion during the sequence of photographs. The article proves that the structure-from-motion technique can be successfully applied to obtain the 3-D cloud top geometry of clouds and should be published after dealing with the following remarks.

**Specific Comments:**

- For the purpose of evaluation, the article yields a case study of tracked features (Fig.2) and an illustration of the retrieved data (Fig.3). While Fig.3 shows that the method allows to detect cloud evolution (arrows), the missing spatial reference, height information and the large dataset makes a proper interpretation difficult.

It would be helpful if the reader would be able to connect the shown 3-D data with the cloud scene shown in in Fig.2. Maybe it is possible to exclude the more distant 3-D data and introduce some regions of interest, such as individual cloud turrets, that could be marked in Fig.2 and then used in Fig.3 to provide a direct connection. Also, the shown arrows could encode the mean height by an appropriate color code, as done in Fig. 5. This might have the advantage that the reader can estimate the cloud geometry directly.

- Fig.5 gives a nice overview of the techniques capabilities on a large scale. Two points of critique here: First, the figure encodes the height as color, but lacks a legend. Second, the figure shows the dataset over a quite large extent. It might help to add a detail view of a specific region of interest contained in the large-scale view, such as a local two-layer situation.

- Fig 6 and 7 may be combined into one figure as both intend to to show (among others) the challenge of a proper comparison between lidar and stereo data.

**Technical Corrections / Suggestions:**

- Page 1, line 5:
"...relatively simple installation on an aircraft...“

Maybe simple in case of a dedicated research aircraft, but most probably not in general.

- Page 1, line 7:
„However we will show that to some extent usable wind information can also be recovered.“

More precise (“to some extent”).

- Page 2, line 21:
„...a big advantage when observing moving and changing clouds.“

Maybe better: „…., so that cloud evolution and motion does not affect the 3-D reconstruction.“

- Page 3, line 16:
„For geometric calibration of the camera we use a common approach.“

Which approach? More precise.

- Page 8, line 6:
„After all filtering...“

Delete „all“.

- Page 8, line 8:
„Such a point cloud is shown in figure 5.“

Maybe just put the figure reference at the end of the previous sentence and delete this sentence („...
relative to a point on the earth's surface (figure 5).“)

- Page 8, line 8-10:
„This point cloud can then be used as a starting point...“

Maybe better: „The point cloud can then serve as reference for other distance measurement
techniques...“ (Which?) „ ...or allow for a 3-D surface reconstruction.“

- Page 9/10, line 20/1:
„Generally, there is a good agreement...“

Maybe better:
„The measured distances between the aircraft and clouds as obtained from the WALES lidar and the
stereo method show a good agreement...“ (typical errors?) „… .The automated comparison between
lidar and the stereo method, however, typically includes a significant number of outliers in multi-
layer cloud situations.“

- Page 10, line 27:
„...have been binned in 1 min bins...“

Maybe better: „….have been binned in time intervals of 1 minute…“

---

## Referee Comment (RC2) · Anonymous Referee #2 · 21 Dec 2018

**Review of "Aircraft based Stereographic Reconstruction of 3D Cloud Geometry" by Kölling et al.**

*Summary:*

This manuscript presents a new method for reconstructing cloud geometry using multiple nadir pictures from an aircraft. This paper also showcases its first application to 2 field campaigns, and a first verification using an active lidar system.

*General Comments:*

This manuscript showcases a novel technique of using well known computer vision techniques to reconstruct cloud geometry and is a valuable contribution to science. This referee suggests this paper should be published, following some revisions, see below for the major and minor issues. The overall content of the paper is well formed, but the introduction and concluding sections require multiple typo corrections. The included comparison to lidar is well received, although the choice of a large area of cloud top height comparison should be revisited.

*Major issues:*

- Verification of the method uses a dubious assumption of cloud homogeneity within 150m of the lidar measurement, refinement should be done, and subsequent conclusions of the lidar representing higher clouds is put into question.
- "cloud surface" has not been defined, yet it underpins this manuscript. Cloud surface is not what the feature selection algorithm is used, but rather cloud surface edges. Clarification should be done.

*Minor issues:*

- Title of the manuscript is slightly misleading, common wording for this methodology is 'Structure-from-Motion', see Westoboy et al., 2012 (amongst others)
- Point selection algorithm choice has not been described. Some description of these selection points, for finding the corners would be a welcomed addition to this manuscript.
- Figure 3 is nearly useless without a better frame of reference. Please include a frame of reference marker. It may be useful to put and 'x-y-z' axis in Fig. 2, and the rotated version of which in Fig. 3.
- Figure 5 should have a colorbar to denote the color scheme of the cloud height.
- Last paragraph of section 3 describe transformation of a point cloud to cartesian 3D, but does reference the use of the aircraft navigation, or potential sources of errors from it.
- Section 4.2 is using data from a status cloud deck to infer cross track stability of the measurement. Further evidence of the status cloud deck's vertical stability should be presented to reinforce this point. If no other is available, is it possible to use a ground target instead of the cloud to cross track stability? Related remarks in the conclusion should be amended
- Last paragraph of the conclusions should be inserted in the methods as well, and references to the appendix.
- A note on the spectral aberrations (if any) would be useful in the appendix A.

*Here are some specific points to be addressed:*

- P.1 line 10, typo: "comparson" should be "comparison"
- P.1 line 16, what the authors describe is unclear: "where observed clouds and observer are at different locations,… "
- P. 1 line 21, why is the term "Finally, …" used at the start of the sentence? Flow of the entire paragraph should be reevaluated.
- P. 1 line 25, "by Ewald (2016); Ewald et al. (2018)" should be "by Ewald (2016) and Ewald et al. (2018)"
- P.1 line 26, Unclear grammar to what "it is shown […]" is referencing, Is it "Ewald et al. (2018) showed that […]" ?
- P. 6 caption of figure 2. Unknown symbol of '^' on top of '=', please define or use more widely known character.
- P. 8 line 3, grammatically unsound "because due to the […]", please rephrase.
- P. 8 line 5, please be more precise in this sentence "these clouds can still be tracked in the presence of sunglint." - related conclusion remarks should also be ammended
- P. 12 line 3, "active remote sensing in the nadir perspective" seems odd, maybe: "nadir pointing active remote sensing"
- P. 12 line 5, please remove capitalization of "Because"
- P. 12 line 7, typo: "requirment" should be "requirement"
- P. 12 line 20, "in stead" should be "instead"

*References:*

Westoby, M. J., Brasington, J., Glasser, N. F., Hambrey, M. J. and Reynolds, J. M.: 'Structure-from-Motion'photogrammetry: A low-cost, effective tool for geoscience applications, Geomorphology, 179, 300–314, doi:10.1016/j.geomorph.2012.08.021, 2012.

---

## Author Response (AR2)

**Introduction**

We thank referee #1 for his/her careful reading, comments and suggestions which we address in the following. The authors' answers are printed in italics.

*Remark: The figure numbers in the referee comments and the page numbers in the authors' answers are corresponding to the original manuscript. If not stated otherwise, figure and equation numbers in the authors' answers are referring to the revised, marked-up manuscript version (showing the changes made) which can be found at the end of this text.*

**General comments**

– The submitted manuscript deals with the 3-D reconstruction of clouds via the structure-from-motion technique using image data obtained from a downward-looking camera installed at a research aircraft. The goal is to provide 3-D cloud top geometry information and geolocation that can be used to improve retrievals by other remote sensing methods, e.g. derivation of cloud droplet radii from spaceborne hyper-spectral imagery. While airborne observation of clouds is a costly enterprise and delivers data only for the flight period, it provides a quite complete and still rare view on the cloud top geometry.

The article describes in detail the methodology and evaluation of the proposed airborne reconstruction, including camera calibration, feature tracking and 3-D reconstruction. Besides an empirical evaluation with an onboard lidar system, the article discusses related challenges of such an approach, such as synchronization with the aircrafts navigation system or the effect of cloud evolution and motion during the sequence of photographs. The article proves that the structure- from-motion technique can be successfully applied to obtain the 3-D cloud top geometry of clouds and should be published after dealing with the following remarks.

→ *Thank you for your helpful and supportive review. We generally agree with your comments and are confident that we could improve the manuscript quite a bit with your support. At the end of this text you will find a diff for the revised manuscript.*

**Specific comments**

– For the purpose of evaluation, the article yields a case study of tracked features (Fig.2) and an illustration of the retrieved data (Fig.3). While Fig.3 shows that the method allows to detect cloud evolution (arrows), the missing spatial reference, height information and the large dataset makes a proper interpretation difficult.

It would be helpful if the reader would be able to connect the shown 3-D data with the cloud scene shown in in Fig.2. Maybe it is possible to exclude the more distant 3-D data and introduce some regions of interest, such as individual cloud turrets, that could be marked in Fig.2 and then used in Fig.3 to provide a direct connection. Also, the shown arrows could encode the mean height by an appropriate color code, as done in Fig. 5. This might have the advantage that the reader can estimate the cloud geometry directly.

→ *Indeed, it is difficult to connect the 3-D data with 2-D images. While it is possible to draw a 2-D image into the 3-D plot, this does not yield much benefit. Showing this data from a different perspective than the camera perspective is almost as hard to understand as the figures presented in the discussion manuscript. Thus we chose to present only a single figure showing the camera's perspective as in the previous figure 2 but added color coded height information as well as cloud movement vectors as you suggested. This way,*

> *the reader can estimate the cloud geometry more easily and relate it to the actual image. Additionally, we marked the location of this image in figure 7, which shows the wind field in the larger area.*

– Fig.5 gives a nice overview of the techniques capabilities on a large scale. Two points of critique here: First, the figure encodes the height as color, but lacks a legend. Second, the figure shows the dataset over a quite large extent. It might help to add a detail view of a specific region of interest contained in the large-scale view, such as a local two-layer situation.

> → *We have added a legend and a magnification of the central part of the scene. It highlights two cloud layers and a small cloud patch above both of them.*

– Fig 6 and 7 may be combined into one figure as both intend to to show (among others) the challenge of a proper comparison between lidar and stereo data.

> → *We have combined the figures and added an arrow to mark the relevant region of the comparison plot.*

**Technical Corrections / Suggestions**

– **P1, 5:** "...relatively simple installation on an aircraft..."

Maybe simple in case of a dedicated research aircraft, but most probably not in general.

> → *We've added "(research)" before "aircraft". We agree that it is certainly easier to install the system on research aircraft. On general purpose aircraft, the lack of apertures might prevent an easy installation but still, the discussed single camera system requires very little space and the only additional requirement is an accurate navigation system, which should be available on most bigger aircraft anyways. If that is not available, such a navigation system mostly consists of a box to be attached statically somewhere on the airframe and connected to a GPS antenna. So compared to other, especially bigger or active sensors, this system is indeed simpler to install on a general aircraft as well.*

– **P1, 7:** „However we will show that to some extent usable wind information can also be recovered."

More precise ("to some extent").

> → *We now refer to the filtering of outliers.*

– **P2, 21:** „...a big advantage when observing moving and changing clouds."

Maybe better: „....., so that cloud evolution and motion does not affect the 3-D reconstruction."

> → *Changed accordingly.*

– **P3, 16:** „For geometric calibration of the camera we use a common approach."

Which approach? More precise.

> → *Has been changed to "For geometric calibration of the camera we use a common approach of analyzing multiple images of a known chessboard pattern to resolve unknown parameters of an analytic distortion model." For further details, the reader is referred to the appendix.*

- **P8, 6:** „After all filtering..."

  Delete „all".

  > → *Changed accordingly.*

- **P8, 8:** „Such a point cloud is shown in figure 5."

  Maybe just put the figure reference at the end of the previous sentence and delete this sentence („... relative to a point on the earth's surface (figure 5).")

  > → *Changed accordingly.*

- **P8, 8-10:** „This point cloud can then be used as a starting point..."

  Maybe better: „The point cloud can then serve as reference for other distance measurement techniques..." (Which?) „ ...or allow for a 3-D surface reconstruction."

  > → *Changed to "This point cloud can be used on its own, serve as a reference for other distance measurement techniques (e.g. oxygen absorption methods (Zinner et al., 2018) distances derived by a method according to Barker et al. (2011)) or allow for a 3D surface reconstruction."*

- **P9/10, 20/1:** „Generally, there is a good agreement..."

  Maybe better: „The measured distances between the aircraft and clouds as obtained from the WALES lidar and the stereo method show a good agreement..." (typical errors?) „.... .The automated comparison between lidar and the stereo method, however, typically includes a significant number of outliers in multi- layer cloud situations."

  > → *We reviewed the data for this comparison in order to better quantify the typical errors. Still, we were not able to find a sensible method of removing clear outliers due to comparing different clouds without manual filtering. We prefer not to introduce an artificial bias into the comparison by adding subjective criteria. Therefore we added a reference to Stevens et al. (accepted) and additional explanation about the difficulties in comparing the sensors (different sensors see different clouds). We decided that out of this reasons, quantitative comparison with lidar data is only useful for bias, not for spread. On the other hand, for homogeneous cloud decks, as investigated in the across track stability section, the internal spread of the stereo method can be quantified (47.3 m standard deviation in this case). We swapped the order of sections 4.1 and 4.2 to support this argument.*

- **P10, 27:** „....have been binned in 1 min bins..."

  Maybe better: „.....have been binned in time intervals of 1 minute..."

  > → *Changed accordingly.*

**References**

Barker, H. W., Jerg, M. P., Wehr, T., Kato, S., Donovan, D. P., and Hogan, R. J.: A 3D cloud-construction algorithm for the EarthCARE satellite mission, Quarterly Journal of the Royal Meteorological Society, 137, 1042–1058, https://doi.org/10.1002/qj.824, http://dx.doi.org/10.1002/qj.824, 2011.

5  Stevens, B., Ament, F., Bony, S., Crewell, S., Gross, S., Hirsch, L., Mayer, B., Wendisch, M., Wirth, M., Bakan, S., Brück, H.-M., Ehrlich, A., Ewald, F., Farrell, D., Forde, M., Gödde, F., Grob, H., Hagen, M., Hansen, A., Jacob, M., Jäkel, E., Jansen, F., Klepp, C., Klingebiel, M., Kölling, T., Konow, H., Mech, M., Peters, G., Rapp, M., Wing, A., and Wolf, K.: A high-altitude long-range aircraft configured as a cloud observatory - the NARVAL expeditions, Bulletin of the American Meteorological Society, accepted.

Zinner, T., Schwarz, U., Kölling, T., Ewald, F., Jäkel, E., Mayer, B., and Wendisch, M.: Cloud geometry from oxygen-A band observations
10  through an aircraft side window, Atmospheric Measurement Techniques Discussions, 2018, 1–20, https://doi.org/10.5194/amt-2018-220, https://www.atmos-meas-tech-discuss.net/amt-2018-220/, 2018.

**Introduction**

We thank referee #2 for his/her careful reading, comments and suggestions which we address in the following. The authors' answers are printed in italics.

*Remark: The figure numbers in the referee comments and the page numbers in the authors' answers are corresponding to the original manuscript. If not stated otherwise, figure and equation numbers in the authors' answers are referring to the revised, marked-up manuscript version (showing the changes made) which can be found at the end of this text.*

**General comments**

– This manuscript showcases a novel technique of using well known computer vision techniques to reconstruct cloud geometry and is a valuable contribution to science. This referee suggests this paper should be published, following some revisions, see below for the major and minor issues. The overall content of the paper is well formed, but the introduction and concluding sections require multiple typo corrections. The included comparison to lidar is well received, although the choice of a large area of cloud top height comparison should be revisited.

→ *Thank you for your very helpful comments. Based on your suggestions, we've had new insights into details of our measurement system and the described method. While the comment on spectral aberration lead to an intensive and interesting re-investigation of our data, we finally decided not to perform any changes as the effects are comparably small.*

**Major issues**

– Verification of the method uses a dubious assumption of cloud homogeneity within 150m of the lidar measurement, refinement should be done, and subsequent conclusions of the lidar representing higher clouds is put into question.

→ *Short version: while the choice of the radius of the comparison cylinder is to some extent arbitrary, no systematic error should be introduced by any particular choice. The differences in cloud representation between stereo and lidar method are expected because of general reasons. Our conclusions merely state that our observations agree with these general reasons.*

*Longer version: the authors are aware that there is no reason, why a cloud should be homogeneous within any chosen radius for a physical reason. This is also not the point to be made in the comparison. The assumption is rather that clouds or parts thereof which are co-located horizontally are also co-located vertically. As discussed later in the corresponding section, this assumption is not valid in areas of multi-layer clouds. In areas of single layer clouds, this assumption should however hold. Comparing data from within a vertical cylinder should therefore result in results of similar height. While the individually measured data pairs may scatter broadly due to cloud inhomogeneities, apart from systematic differences in the measurement principle, there is no reason to believe that the mean or median deviation of all data pairs should be different from zero.*

*The size of the cylinder is rather arbitrary but the particular choice has reasons: the aircraft moves at a speed of approximately $200$ m/s and the data of the lidar system is available at $1$ Hz and averaged over this period. Any comparison between both systems should therefore be in the order of $200$ m horizontal resolution. Furthermore, data derived from the stereo method is only available where the method is confident that it worked. Thus not every lidar data point has a corresponding stereo data point. Increasing*

*the size of the cylinder increases the count of data pairs, but also increases false correspondences. The general picture however remains unchanged.*

*The statement that lidar is representing higher (parts of) clouds is not really a conclusion but rather an expectation due to the measurement principles. First, the stereo method is using slanted observation directions in addition to the nadir direction, thus and because it uses regions of high image contrast, the method tends to favour cloud sides, which are below the top of a cloud. The lidar observes any part of the cloud which is visible from directly above. On average, the parts of the cloud which are observed by stereo thus should be below of the parts which are observed by the lidar method. Second, the lidar has been designed to be able to observe barely visible or even invisible parts of the atmosphere, so it is clearly more sensitive than the camera. As the algorithms used select signals closest to the aircraft, a more sensitive system should prefer higher clouds.*

*Our conclusion states that our analysis is in accordance to this expectation.*

- "cloud surface" has not been defined, yet it underpins this manuscript. Cloud surface is not what the feature selection algorithm is used, but rather cloud surface edges. Clarification should be done.

  → *We have added a paragraph describing our definition of the cloud surface to the introduction. We also changed the following paragraph, which now reflects that we are searching for points on the surface, rather than generating a full surface model.*

**5 Minor issues**

- Title of the manuscript is slightly misleading, common wording for this methodology is 'Structure-from-Motion', see Westoboy et al., 2012 (amongst others)

  → *We have thought about this potentially misleading naming and think that for our method, stereo is slightly more applicable as Structure-from-Motion (SfM). As Westoby et al. (2012) describe, a typical feature of SfM methods is to reconstruct both, the observed structure as well as the motion of the camera only from an image sequence. Due to the unknown movement and deformation of the observed clouds, this approach is less feasible in our case. Also our own experiments show that the commonly and particularly used SIFT feature matching algorithm (Lowe, 2004) does not work very good on non-solid surfaces as clouds and ocean provide. Due to these limitations, we depend on highly accurate position and orientation information of the aircraft, which is in our case provided by the aircraft's basis instrumentation. Furthermore, SfM methods tend to use more than two images simultaneously to perform object recognition. This also does not happen in our method, but rather the results of processed image pairs are combined in a later step. Thus the actual image processing step is very close to classical stereographic methods. So while it is true that we uses the camera's motion to derive structural information, we do not use typical SfM techniques. This is why we choose not to use the term Structure-from-Motion.*

- Point selection algorithm choice has not been described. Some description of these selection points, for finding the corners would be a welcomed addition to this manuscript.

  → *We have reformulated the corresponding paragraph and explain the point selection algorithm and the reasoning behind choosing this algorithm in more detail.*

– Figure 3 is nearly useless without a better frame of reference. Please include a frame of reference marker. It may be useful to put and 'x-y-z' axis in Fig. 2, and the rotated version of which in Fig. 3.

> → *This has also been pointed out by referee #1. We agree that this figure is barely understandable. Displaying the corresponding data from a different than the camera's perspective remains to be hard to understand, so we decided to combine figures 2 and 3 into an improved new figure 2. This way, the points can be associated visually to the image and height information is now included as color codes.*

– Figure 5 should have a colorbar to denote the color scheme of the cloud height.

> → *Changed accordingly*

– Last paragraph of section 3 describe transformation of a point cloud to cartesian 3D, but does reference the use of the aircraft navigation, or potential sources of errors from it.

> → *In the beginning of section 3, we explain that the transformation into a geocentric reference frame is performed using the aircraft's navigation system. Together with the addition about landmarks (see next item), our procedure should now be more clear to the reader. By adding the paragraph you proposed in the second next item to the end of section 3, the reader is pointed to the requirement of having an accurate time synchronization (at the order of tens of milliseconds) between navigation system and sensors, which we see as the biggest challenge in transforming the points to a geocentric reference frame.*

– Section 4.2 is using data from a status cloud deck to infer cross track stability of the measurement. Further evidence of
10 the status cloud deck's vertical stability should be presented to reinforce this point. If no other is available, is it possible to use a ground target instead of the cloud to cross track stability? Related remarks in the conclusion should be amended

> → *In fact, the orientation of the camera with respect to the aircraft has been determined independently of any clouds by aligning camera images of landmarks (taken on multiple flights) with satellite images. We added an according note to the first paragraph of chapter 3. In consequence, the orientation of the cloud deck used to determine the across track stability of the presented method has not been used to determine the camera's orientation. We thus are quite confident that potential systematic deviations would be visible in the presented plot. Remaining inhomogeneities should already be included as detrimental effects in our conclusions. We avoid using ground targets to assess across track stability, first because we want to include effects which are specific to observing clouds (if any) and second, because most of our measurements have been conducted over ocean or clouds, leaving only a few ground targets, which also have been used to align the camera and thus should not be used for assessment to avoid circular logic. For further clarification, a note has been added to the stability analysis as well.*

– Last paragraph of the conclusions should be inserted in the methods as well, and references to the appendix.

> → *A modified version of the last paragraph of the conclusions has been inserted at the end of the methods section.*

15 – A note on the spectral aberrations (if any) would be useful in the appendix A.

*→ It is indeed a very good idea to look at spectral aberrations, which we did not prior to your comment. To further investigate effects of spectral aberration, we rerun our calibration procedure for all color channels separately and reprocessed parts of the measurement data. A comparison of different calibration data reveals that choosing a different calibration changes an observed viewing angle difference in the order of $1 - 2$‰. For a cloud distance of $10$ km as in the calibration example, this translates to uncertainties of about $10$ m, which is also confirmed by rerunning the analysis of the horizontal cloud deck. Interestingly, despite the same order of magnitude, we did not find that the curvature of the cloud deck analysis can be removed by considering spectral aberrations. Due to effectively using fewer pixels when doing the camera calibration procedure on a single channel image, the reprojection error is increased accordingly. In the end, we choose not to use different calibration data for different color channels, as the effects turn out to be smaller than other sources of error and using only a single calibration facilitates data handling. We did not consider effects of spectral aberrations within a single color channel, but assume that these effects should be even smaller and thus can be neglected as well. We have added a note on this analysis to appendix A.*

**Here are some specific points to be addressed**

- **P.1 line 10:** typo: "comparson" should be "comparison"

  *→ Changed accordingly*

- **P.1 line 16:** what the authors describe is unclear: "where observed clouds and observer are at different locations,... "

  *→ The sentence has been reworded.*

- **P. 1 line 21:** why is the term "Finally, ..." used at the start of the sentence? Flow of the entire paragraph should be reevaluated.

  *→ The sentence starts an additional use case. We have split the paragraph into two changed the wording.*

- **P. 1 line 25:** "by Ewald (2016); Ewald et al. (2018)" should be "by Ewald (2016) and Ewald et al. (2018)"

  *→ Changed accordingly*

- **P.1 line 26:** Unclear grammar to what "it is shown [...]" is referencing, Is it "Ewald et al. (2018) showed that [...]" ?

  *→ Changed to "In particular, Ewald (2016) and Ewald et al. (2018) have shown ...".*

- **P. 6 caption of figure 2:** Unknown symbol of '^' on top of '=', please define or use more widely known character.

  *→ Due to reworking the previous figures 2 and 3, the mentioned symbol is not present anymore.*

- **P. 8 line 3:** grammatically unsound "because due to the [...]", please rephrase.

  *→ Has been rephrased.*

- **P. 8 line 5:** please be more precise in this sentence "these clouds can still be tracked in the presence of sunglint." - related conclusion remarks should also be ammended

> → *We now describe the situation more precisely, both in the marked section, as well as in the respective part of the conclusions.*

– **P. 12 line 3:** "active remote sensing in the nadir perspective" seems odd, maybe: "nadir pointing active remote sensing"

> → *Changed accordingly*

– **P. 12 line 5:** please remove capitalization of "Because"

> → *Changed accordingly*

– **P. 12 line 7:** typo: "requirment" should be "requirement"

> → *Changed accordingly*

– **P. 12 line 20:** "in stead" should be "instead"

> → *Changed accordingly*

[revised manuscript text omitted]